# The Type I Interferon Axis in Systemic Autoimmune Diseases: From Molecular Pathways to Targeted Therapy

**DOI:** 10.3390/biom15111586

**Published:** 2025-11-12

**Authors:** Ryuhei Ishihara, Ryu Watanabe, Mayu Shiomi, Yuya Fujita, Masao Katsushima, Kazuo Fukumoto, Shinsuke Yamada, Motomu Hashimoto

**Affiliations:** 1Department of Clinical Immunology, Osaka Metropolitan University Graduate School of Medicine, 1-4-3, Asahi-machi, Abeno-ku, Osaka 545-8585, Japan; 2Department of Rheumatology, Hyogo Prefectural Amagasaki General Medical Center, Amagasaki 660-8550, Japan

**Keywords:** interferons, IFN signature, JAK–STAT, SLE, RA, vasculitis, stratified medicine

## Abstract

Type I interferons (IFN-I) are pivotal effectors of innate immunity and constitute a central axis of host defense against pathogens. Sensing of exogenous or endogenous nucleic acids by pattern-recognition receptors—exemplified by Toll-like receptors—triggers transcriptional induction of IFN-I. Engagement of the heterodimeric IFN-I receptor on nucleated cells reprograms cellular states via canonical Janus kinase–signal transducer and activator of transcription (JAK–STAT) signaling as well as STAT-independent, noncanonical pathways. This axis is tempered by multilayered regulatory mechanisms, including epigenetic remodeling, and important aspects remain incompletely defined. Dysregulation of IFN-I activity underlies diverse autoimmune disorders, notably systemic lupus erythematosus, wherein IFN-responsive gene signatures stratify disease endotypes, reflect disease activity trajectories, and predict therapeutic responsiveness. In recent years, therapeutic strategies targeting this pathway are now available: anti-IFN-I receptor therapy for systemic lupus erythematosus (SLE) and JAK inhibition for rheumatoid arthritis (RA) and giant cell arteritis (GCA). Altogether, a refined understanding of the IFN-I axis furnishes a pragmatic framework for patient stratification, response prediction, and mechanism-informed therapy design across immune-mediated diseases.

## 1. Introduction

Type I interferons (IFN-I) constitute a cytokine family encompassing IFN-α and IFN-β that orchestrate innate immunity by governing antiviral responses and shaping the activation of adaptive immunity [1,2]. IFN was first identified as a factor produced by influenza virus-infected chick embryo cells that conferred resistance to infection by both homologous and heterologous viruses [3]. Early reports characterized IFN as a wholly non-toxic antiviral with no effects in uninfected cells; however, by the mid-1970s, evidence had already emerged indicating that IFN can exert deleterious effects in vivo [4]. Contemporaneously, Skurkovich et al. demonstrated that administration of anti-IFN serum reduces both cell-mediated and humoral immunity in mice [5]. They subsequently hypothesized that IFN is a central driver of immune dysregulation across diverse autoimmune diseases, advocated the therapeutic potential of IFN neutralization, and established a conceptual framework for IFN biology and its clinical targeting in autoimmunity [6,7].

Over the past five decades, converging evidence has implicated dysregulated IFN-I activity as a principal pathogenic driver in systemic autoimmune diseases, including systemic lupus erythematosus (SLE), rheumatoid arthritis (RA), and various forms of vasculitis [8,9,10,11]. Among these conditions, SLE is characteristically marked by a persistent “type I interferon signature,” defined by sustained upregulation of IFN-stimulated genes in peripheral blood and affected tissues [12]. In RA, a subset of patients displays an interferon-inducible molecular phenotype that prognosticates therapeutic response and disease trajectory [13]. Similarly, in giant cell arteritis and other primary vasculitis, recent transcriptomic analyses and serological evidence substantiate a contributory role for IFN-I in vascular inflammation and tissue injury [14,15]. Moreover, dysregulation of the type I interferon system has been documented in several other autoimmune rheumatic diseases, including primary Sjögren’s syndrome, systemic sclerosis, and dermatomyositis [16,17,18]. Although the reasons why elevated IFN-I activity is observed across these clinically divergent conditions remain incompletely defined, they can be rationalized within a unifying framework in which heterogeneous upstream triggers converge on a conserved axis that begins with the sensing of nucleic acids and proceeds through the IFN-I receptor (IFNAR) and the Janus kinase–signal transducer and activator of transcription (JAK–STAT) pathway. This conceptual view has, in turn, catalyzed efforts to therapeutically modulate the interferon pathway. Notably, the recent approval of a monoclonal antibody targeting IFNAR in SLE [19], together with the expanding deployment of JAK–STAT inhibitors that attenuate signaling downstream of IFN-I [20], marks a substantive advance toward precision medicine by enabling patient stratification and individualized therapeutic selection. In autoimmune diseases, JAK inhibitors, originally developed for rheumatoid arthritis, are now being investigated across a wide range of autoimmune disorders, underscoring both their robust therapeutic efficacy and the common interferon-driven pathogenic mechanisms [21]. This review delineates the IFN-I signaling axis, elucidates the contributions of IFN-I to the pathogenesis of RA, SLE, and vasculitis, and surveys therapeutic strategies that directly or indirectly target the IFN-I system.

## 2. Type I Interferon Biology and Signaling Architecture

### 2.1. Molecular Diversity and Subtype Characteristics

IFN-I are pivotal initiators of innate immune defenses against a broad spectrum of pathogens [22]. The IFN-I family comprises IFN-α, IFN-β, IFN-δ, IFN-ε, IFN-κ, IFN-ω, and IFN-ζ, with IFN-α represented by 13 distinct homologs in humans and 14 in mice. Among these subtypes, humans express IFN-α, IFN-β, IFN-ε, IFN-κ, and IFN-ω [23].

IFN-I, particularly IFN-β, can be produced by most cell types; however, antiviral IFN-I production by blood cells predominantly relies on nucleic acid sensing via Toll-like receptor (TLR)7 and TLR9 in plasmacytoid dendritic cells (pDCs) [24,25]. Additionally, tissue macrophages, epithelial cells, and fibroblasts contribute to IFN-I production in a context-dependent manner [26,27].

IFN-I are constitutively expressed under physiological conditions, sustaining tissue homeostasis and maintaining a poised antiviral state [28]. Upon detection of pathogen-associated molecular patterns (PAMPs) derived from viral or bacterial nucleic acids, or damage-associated molecular patterns (DAMPs) originating from tissue injury, by pattern-recognition receptors (PRRs), diverse cell types—including pDCs—induce IFN-I production [29]. The response unfolds in two phases: an initial, IFN regulatory factor 3 (IRF3)-dominant wave, followed by IRF7 activation that establishes a positive feedback loop and amplifies IFN-I secretion [23,30,31]. Secreted IFN-I amplifies IFNAR signaling through autocrine and paracrine mechanisms, thereby promoting antigen presentation, chemokine production, and the coordinated reprogramming of innate and adaptive immunity via broad induction of interferon-stimulated genes (ISGs) [29,32,33]. Conversely, negative regulators such as suppressor of cytokine signaling 1 (SOCS1) and ubiquitin-specific protease 18 (USP18) [34], together with cytokine milieu-dependent thresholds [32,35], finely modulate the magnitude and duration of this response.

### 2.2. Upstream Sensing Pathways: TLRs, RLRs, and cGAS–STING

The initiation of IFN-I production is governed by PRRs that sense exogenous or endogenous nucleic acids [36]. These include plasma membrane TLRs such as TLR4, endosomal TLRs such as TLR7 and TLR9, and cytosolic sensors such as retinoic acid-inducible gene I (RIG-I) and melanoma differentiation-associated gene 5 (MDA5) [37,38,39,40] (Figure 1). These pathways can also be engaged by endogenous ligands released from injured or apoptotic cells, indicating that the detection of self-nucleic acids is pivotal in sterile inflammation and autoimmunity [41].

Ten TLRs have been identified in humans. Cell surface TLRs (TLR1, TLR2, TLR4, TLR5, TLR6, TLR10) primarily detect microbial membrane components and elicit proinflammatory responses, whereas intracellular TLRs (TLR3, TLR7, TLR8, TLR9) principally recognize microbial nucleic acids of bacterial or viral origin, triggering both IFN-I production and inflammatory signaling [42].

The cell surface receptor TLR4, a founding member of the Toll-like receptor family, senses bacterial lipopolysaccharide (LPS), a constituent of the outer membrane of Gram-negative bacteria [43]. TLR2 participates in the recognition of a broad spectrum of PAMPs derived from bacteria, fungi, parasites, and viruses, typically forming heterodimers with TLR1 or TLR6 [43,44,45]. TLR5 recognizes flagellin, the principal protein component of bacterial flagella [43]. Non-TLR membrane receptors implicated in these pathways include members of the tumor necrosis factor (TNF) receptor superfamily and the receptor activator of nuclear factor κB (RANK) [46,47,48]. Both TNF and receptor activator of NF-κB ligand (RANKL) have been implicated in promoting sustained expression of ISGs through paracrine and autocrine signaling mechanisms [32,41].

By contrast, intracellular TLRs are predominantly localized to endosomes [41]. TLR3 senses double-stranded ribonucleic acid (RNA), whereas TLR7 and TLR8 detect single-stranded RNA [49,50]. TLR9 recognizes unmethylated cytosine–phosphate–guanine (CpG) motifs within deoxyribonucleic acid (DNA) [51].

TLR signaling is broadly categorized into two pathways: the myeloid differentiation primary response 88 (MyD88)-dependent pathway and the MyD88-independent pathway [42]. MyD88, the first identified Toll/interleukin-1 receptor (TIR) domain-containing adaptor protein is employed by all TLRs except TLR3 and principally activates NF-κB signaling [42]. Upon ligand engagement, MyD88 associates with the receptor via its TIR domain, recruits the serine/threonine interleukin (IL)-1 receptor-associated kinases (IRAKs), and nucleates a signaling complex with IRAK family members known as the myddosome [52,53,54]. Activated IRAKs engage TNF receptor-associated factor 6 (TRAF6), leading to activation of the IκB kinase complex [55]. The inhibitor of κB kinase (IKK) complex phosphorylates IκB, triggering its degradation and permitting the transcription factor NF-κB to translocate into the nucleus [56].

By contrast, TLR3 and TLR4 signal through a MyD88-independent pathway. Upon activation, TIR domain-containing adaptor protein-inducing IFN-β (TRIF) engages these receptors via its TIR domain, promoting TRAF3-dependent activation of the IKK-related kinase TANK-binding kinase 1 (TBK1). TRAF3 activates TBK1 and IKK—often within complexes containing NF-κB essential modulator (NEMO)—culminating in IRF3 phosphorylation and dimerization. The IRF3 homodimer then translocates to the nucleus, where it drives transcription of IFN-I genes and ISGs [57,58]. The MyD88-independent arm of TLR4 signaling additionally requires the TRIF-related adaptor molecule (TRAM) [59].

Following the discovery of TLRs, multiple classes of cytosolic PRRs have been delineated, notably the RIG-I–like receptors (RLRs) and the nucleotide-binding oligomerization domain (NOD)-like receptors (NLRs) [42].

The RLR family comprises three members— RIG-I, MDA5, and LGP2 (protein named “laboratory of genetics and physiology 2”)—which serve as the principal cytosolic sensors of RNA and detect RNA viruses [60,61,62,63]. In addition, RIG-I and MDA5 contain two N-terminal caspase activation and recruitment domains (CARDs) that mediate downstream signaling. Downstream signaling is transmitted via mitochondrial antiviral signaling protein (MAVS) to TBK1 and IKKε, which in turn activate IRF3 and IRF7; together with NF-κB, these factors drive transcription of IFN-I and ancillary antiviral genes [64]. Moreover, AT-rich DNA can be transcribed by RNA polymerase III into 5′-triphosphorylated RNA that functions as a RIG-I agonist [23].

Cyclic GMP–AMP synthase (cGAS) is activated upon binding double-stranded DNA (dsDNA), catalyzing the conversion of adenosine triphosphate (ATP) and guanosine triphosphate (GTP) into cyclic GMP–AMP (cGAMP). cGAMP, together with other cyclic dinucleotides (CDNs), engages STING on the endoplasmic reticulum (ER). Following activation, STING translocates from the ER to the ER–Golgi intermediate compartment (ERGIC) and recruits TBK1. TBK1-mediated phosphorylation of STING and IRF3 drives IRF3 nuclear translocation, thereby inducing IFN-I production [59,65].

Additional cytosolic DNA motifs are sensed by diverse receptors—including DNA-dependent activators of interferon regulatory factors (DAIs) and DEAD- and DEAH-box (DExD/H-box) helicases—all of which have been implicated in IFN-I induction [61,66,67]. Moreover, the cytosolic pattern-recognition receptor NOD1 initiates IFN-I signaling upon sensing γ-D-glutamyl–meso-diaminopimelic acid (iE-DAP), a peptidoglycan motif characteristic of Gram-negative and select Gram-positive bacteria. By contrast, NOD2 recognizes muramyl dipeptide (MDP), a conserved peptidoglycan fragment present in both Gram-positive and Gram-negative bacteria, and also responds to viral ssRNA [68,69,70,71]. These receptors recruit receptor-interacting serine/threonine kinase (RICK) via CARDs. RICK activates the IKK complex through NEMO, enabling NF-κB nuclear translocation [72].

Thus, induced phosphorylated IRF3/IRF7 and NF-κB bind to the promoter regions of the IFN-I gene family, thereby regulating IFN-I expression [73,74].

### 2.3. Canonical IFNAR–JAK–STAT Signaling Cascade

All nucleated cells express functional transmembrane IFN-I receptors, which are generally composed of IFNAR1 and IFNAR2 heterodimers [75,76].

IFN-I signals through IFNAR1/IFNAR2, which couple to TYK2 and JAK1, respectively; ligand engagement induces receptor rearrangement and JAK activation, generating phosphotyrosine docking sites for STATs [23] (Figure 2). Phosphorylated STAT1 and STAT2 assemble with IRF9 to form interferon-stimulated gene factor 3 (ISGF3), which translocates to the nucleus, binds interferon-stimulated response elements (ISREs), and induces ISGs—for example, myxovirus resistance 1 (MX1), 2′–5′-oligoadenylate synthetase 1 (OAS1), and members of the interferon-induced proteins with tetratricopeptide repeats (IFIT) family—thereby establishing antiviral and immunoregulatory programs [77,78]. In specific contexts, STAT1 or STAT3 homodimers engage gamma-activated sequence (GAS) elements to drive distinct ISG modules [79,80]; unphosphorylated STATs also contribute to transcriptional regulation [81].

### 2.4. Noncanonical Signaling Routes and Crosstalk

Beyond the canonical JAK–STAT cascade, IFNAR1/IFNAR2 engagement activates STAT-independent PI3K and MAPK pathways, thereby broadening IFN-I effector outputs [77] (Figure 2). In the PI3K–AKT axis, IFN-I–induced phosphorylation of insulin receptor substrate 1 (IRS1) recruits the p85 regulatory subunit of PI3K and activates the p110 catalytic subunit [82]. PI3K converts phosphatidylinositol 4,5-bisphosphate (PIP2) into phosphatidylinositol 3,4,5-trisphosphate (PIP3) [83], which co-localizes 3-phosphoinositide–dependent protein kinase 1 (PDK1) and AKT at the plasma membrane; PDK1 subsequently phosphorylates AKT at Thr308 [84,85], propagating signaling to the mechanistic target of rapamycin (mTOR), a central regulator of protein translation and cellular metabolism [86]. Within the MAPK arm, JAK phosphorylates VAV and other guanine-nucleotide-exchange factors (GEFs), leading to RAC1 activation and sequential activation of MAPK kinase kinase (MAPKKK) and MAPK kinase (MAPKK), most prominently MKK3/6, which in turn activate p38 MAPK [87,88]. Activated p38 signals to MAPK-activated protein kinase (MAPKAPK)2/3, mitogen- and stress-activated kinase 1 (MSK1), and MAPK-interacting protein kinase 1 (MNK1) [89,90]; in mammals, ERK and c-Jun N-terminal kinase (JNK) can also contribute [78].

### 2.5. Regulatory Checkpoints of IFN-I Activity

Although the IFN-I response is highly potent, excessive or sustained activation can precipitate autoimmunity and tissue injury. Accordingly, cells finely calibrate its amplitude, duration, and gene selectivity through multiple regulatory layers [91,92]. Proximally, receptor- and JAK-level feedback, phosphatase activity, and post-translational modifications limit signaling strength and timing. At the epigenetic and RNA-mediated layer, chromatin remodeling, histone/DNA modifications, and non-coding RNAs shape ISG accessibility, transcriptional elongation, and persistence. We summarize these mechanisms and their net effects in Table 1. IFN-induced epigenetic alterations can persist for days to weeks as transcriptional memory, sustaining ISG expression after upstream JAK–STAT activity has waned. Even non-ISGs undergo bookmarking, which reprograms responsiveness to subsequent stimuli [93,94].

## 3. Functional Landscape of Type I Interferon Across the Immune Systems

IFN-I reprogram functional states across multiple compartments of innate and adaptive immunity, with an early antiviral program at the core during initial infection. Their effects are bidirectional—potentiating or suppressive—depending on phase (acute versus chronic), concentration, receptor composition, cytokine milieu, and tissue niche; even within a single cell, outcomes can invert according to context.

### 3.1. B Cells

IFN-I modulate B-cell survival, differentiation, and antibody production in a context-dependent fashion. Notably, strain background influences these effects: in C57BL/6 mice, Ifnar1 deficiency yields minimal changes in peritoneal and splenic B-cell subsets, whereas in Ifnar1-deficient mice on the 129Sv background, alterations in the immature bone marrow B-cell repertoire have been reported, accompanied by differential susceptibility to B cell receptor (BCR)-dependent blockade of terminal differentiation under identical conditions [132]. Functionally, IL-6 costimulation promotes plasma-cell differentiation [133]. By contrast, reports on survival and proliferation are inconsistent, describing both enhancement and inhibition [134], a discrepancy sometimes attributed to dose dependence, whereby low IFN-I doses augment immunoglobulin production while high doses suppress it [135].

### 3.2. T Cells

The canonical pathway of Th1 differentiation entails IL-12 engagement of its receptor, activation of JAK2 and TYK2, phosphorylation of STAT4, and consequent induction of T-bet [136]. By contrast, IFN-α/β can acutely phosphorylate STAT4, but in human CD4^+^ T cells this activation is transient and insufficient to drive robust Th1 differentiation or IFN-γ production [137]. Nevertheless, in vivo evidence indicates that IFN-α/β contributes to Th1 establishment in pathogen-responsive contexts [138]. Moreover, Th1 cells can arise in IL-12-deficient settings, implicating alternative or cooperative pathways [139].

Th2 differentiation is driven primarily by IL-4, with the transcription factor GATA3 promoting transcription across the IL-4/IL-5/IL-13 locus while concurrently repressing IL-12Rβ2 to inhibit Th1 commitment. IL-33 and thymic stromal lymphopoietin (TSLP) function as costimulatory inputs, and GATA3 sustains Th2 identity through a self-reinforcing circuit [140,141]. In contrast, IFN-I exerts suppressive effects on the Th2 axis: early studies demonstrated inhibition of IL-5 secretion and eosinophil recruitment in allergic inflammation [142,143]. More recent work indicates that, in humans, IFN-α/β robustly and selectively inhibits IL-4–dependent Th2 differentiation and destabilizes committed Th2 cells, whereas in mice it does not exert comparably strong inhibition [144].

Th17 cells are induced by cooperative signaling from transforming growth factor-β (TGF-β), IL-6, IL-23, and IL-1β, which activates RORγt and drives production of IL-17A, IL-17F, and IL-22, thereby contributing to diverse inflammatory processes [145]. IFN-I suppress Th17 differentiation in mice [146], and this inhibition extends to human Th17 cells [147]. Clinically, although IFN-β therapy is employed in multiple sclerosis, certain patient subsets exhibit limited efficacy or worsening of Th17-dominant inflammation [148]. A contributory mechanism involves IFN-β–induced C-C motif chemokine ligand (CCL)2 production [149], which recruits Th17.1 cells and inflammatory monocytes into the central nervous system; these monocytes upregulate IL-1β and locally differentiate into dendritic cells, thereby further amplifying Th17 responses [150,151].

### 3.3. DCs

DCs initiate antiviral T-cell responses [152]. IFN-I promotes DC maturation, augmenting major histocompatibility complex (MHC) class II and costimulatory molecule expression and enhancing antigen-presenting capacity; they also drive differentiation from plasmacytoid to myeloid DCs, thereby strengthening T-cell activation [152,153]. Conversely, during persistent IFN-I signatures, myeloid differentiation and DC proliferation are suppressed, yielding an inhibitory phenotype characterized by reduced splenic CD11c^+^ cells and elevated PD-L1 and IL-10 expression [154,155,156]. Moreover, whereas TLR stimulation rapidly diminishes CIITA and de novo MHC-II synthesis alongside antigen processing during maturation, IFN-I elicit a distinct maturation program that preserves CIITA and MHC-II expression while simultaneously increasing surface MHC-II and sustaining intracellular antigen processing [152,153].

### 3.4. Monocytes, Macrophages, and NK Cells

Inflammatory monocytes are rapidly mobilized to sites of infection, where they augment the antiviral functions of local and neighboring immune cells, amplify inflammation, and differentiate into macrophages and DCs [157]. Conversely, IFN-I have also been reported to suppress the activity of proinflammatory NOS2^+^ Ly6C^−^ monocytes, modulating both the magnitude and quality of the monocyte response through dual mechanisms of enhanced recruitment and functional inhibition [158].

IFN-I generally exerts inhibitory effects on macrophages. Notably, they downregulate IFN-γ receptor expression and attenuate cellular sensitivity to IFN-γ stimulation [159]. Accordingly, in certain bacterial infections, such as tuberculosis, IFN-I signaling has repeatedly been implicated in adverse host outcomes, in part due to impaired macrophage activation arising from dampened IFN-γ-dependent pathways [160,161,162].

IFN-I orchestrate the activation and regulation of natural killer (NK)-cell function. The IFNAR–STAT axis underpins cytotoxicity and IFN-γ production; STAT1-deficient mice display reduced NK-cell cytotoxicity and heightened virus-induced mortality [154]. Conversely, IFN-I acts as a biphasic regulator: during chronic lymphocytic choriomeningitis virus (LCMV) infection, IFNAR blockade restores NK-cell IFN-γ production, whereas sustained IFN-I drives an inhibitory phenotype via PD-L1 and IL-10 induction [154]. Clinically, pegylated IFN-α2 administration enhances NK-cell activation and TRAIL/CD107a expression, while potentially diminishing IFN-γ–positive NK cells [163,164]. A timing-dependent effect is also evident: early responses are dominated by STAT4-mediated IFN-γ induction, but with prolonged IFN-I exposure the balance shifts toward STAT1-dependent suppression, ultimately curtailing IFN-γ production—a transcription-factor switch that has been proposed mechanistically [165,166,167].

## 4. Type I Interferon in Systemic Lupus Erythematosus

SLE is a chronic, multisystem inflammatory disorder that manifests in distinct clinical trajectories, including chronic persistent, relapsing–remitting, and long-term remission phenotypes [168,169]. Its pathogenesis centers on aberrant immune activation arising from impaired clearance of apoptotic cells, loss of self-tolerance, T- and B-cell dysfunction, and dysregulated cytokine networks [170]. The IFN-I pathway contributes to SLE susceptibility through both genetic predisposition and perturbations of innate immune regulation [171]. Although not a singular etiologic driver in all patients, IFN-I signaling is thought to underlie a substantial component of the disease’s pathophysiological heterogeneity [172].

Both clinical and experimental data implicate IFN-I in disease initiation. De novo SLE has been reported in patients receiving recombinant IFN-α therapy, with symptom improvement following discontinuation [11,173]. Elevated circulating interferon activity is frequently observed in unaffected relatives and exhibits familial correlation [174,175], while longitudinal analyses reveal a steep increase in interferon activity approximately one year before diagnosis [176]. Historically, exogenous IFN induction accelerated disease in NZB/NZW F1 mice [8]. In humans, hundreds of ISGs are markedly upregulated in peripheral blood, establishing IFN-I signaling as the most prominently activated molecular pathway in SLE [177,178].

### 4.1. Biomarker Landscape: Blood and Tissue Interferon Signatures

In SLE, IFN-α is the predominant IFN-I; however, its activity varies widely among patients, with serum IFN-α within the normal range in 40–50% of cases [174]. Thus, IFN-I are unlikely to be a singular causal driver in all patients and may instead constitute a key determinant of the disease’s heterogeneity [172]. Recent studies indicate that elevated baseline IFN-α levels correlate with adverse renal outcomes, including recurrent nephropathy, and may serve as a biomarker for identifying high-risk individuals preemptively [179]. In cohorts with heightened IFN-I activity, strong associations are observed with autoantibodies such as anti-dsDNA and anti-Ro, and interferon signatures are detected across ancestral populations [172]. By contrast, in African Americans, the signature has been reported to depend on anti-RNP antibodies [172,180]. Serum IFN-α levels also exhibit an inverse correlation with complement C3/C4 [181].

Furthermore, peripheral blood interferon signatures correlate with disease severity and organ involvement. Cross-sectional analyses link these signatures to the number of diagnostic criteria fulfilled and to renal, central nervous system (CNS), and hematologic manifestations [182]. Longitudinally, however, interferon signatures are relatively stable and exhibit limited predictive power for relapse [183,184]. By contrast, interferon- and cytokine-inducible chemokines such as C-X-C motif chemokine ligand (CXCL)10, CCL2, and CCL19 correlate with disease activity and may aid in forecasting long-term relapse risk [185]. At the transcriptomic level, non–interferon modules—most notably the plasmablastic signature—often track disease activity more closely [186], suggesting that IFN-I may exert its greatest influence during the early phase of disease around onset.

Concordant findings emerge from tissue and single-cell analyses. Nearly all peripheral immune cell populations in SLE display IFN-I response, with particularly pronounced activation in monocytes [187]. Robust expression of IFN-inducible genes is also observed in synovial tissue from patients with arthritis and in renal biopsies from class IV lupus nephritis [188,189]. Single-cell RNA-seq of kidney biopsies detects interferon signatures across multiple subsets of infiltrating leukocytes and resident tissue cells [190,191] (Figure 3) (Box 1).

Box 1Single-cell reanalysis of AMP RA/SLE datasets.Analyses were performed in R 4.4.2 using Seurat 5.3.0. Raw UMI matrices were imported as sparse objects; cell metadata were merged; where available, CITE-seq ADT data were linked per cell and included as a separate assay. RNA counts were normalized, variable features selected (VST, 2000 genes), scaled, and reduced by PCA; samples were integrated with Seurat’s CCA-based workflow (IntegrateLayers; k.weight = 30), followed by graph construction, Leiden clustering (resolution = 0.5), and UMAP (dims 1–30). ADT was CLR-normalized. Cluster identities were assigned by canonical RNA markers with ADT support. IFN-stimulated genes (ISG) activity was quantified per cell using a six-gene signature (IFI27, IFI44L, IFIT1, ISG15, RSAD2, SIGLEC1); scoring used UCell.

### 4.2. Genetics, Pathways, and Epigenomic Imprinting of the IFN Program

SLE susceptibility is linked to single-nucleotide polymorphisms (SNPs) within the IFN-I axis, including IRF5/7/8, STAT4, and TYK2 [192,193]. The IRF family constitutes a core set of transcription factors that regulate IFN-I and ISGs transcription in a cell type-specific manner [194]. IRF5, in particular, governs inflammatory cytokine production and IFN-I induction in innate immune cells and modulates B-cell responses downstream of TLR stimulation [194]. In SLE, IRF5 risk alleles associate with elevated circulating IFN-I activity in a manner contingent upon anti-RNA-binding protein (anti-RBP) or anti-dsDNA autoantibodies [195,196]. Moreover, the IRF5 risk haplotype correlates with autoantibody production even in healthy individuals, suggesting a potential feedforward loop encompassing enhanced B-cell TLR signaling, nucleic acid-containing immune complex formation, innate-cell TLR activation, and excessive interferon production [197].

Genetic variants in IRF7 and IRF8 likewise confer increased SLE risk and are associated with altered IFN-I responses in patients [194]. Beyond the IRF family, numerous genes that modulate interferon pathway function—including STAT4, MAVS, IFIH1, and PTPN22—also contribute to susceptibility, underscoring IFN activity in SLE as a polygenic trait [198]. Moreover, genome-wide association studies (GWAS) stratified by circulating IFN-α levels have identified loci such as PRKG1, PNP, and ANKS1A—difficult to detect in conventional case–control designs—as associated with elevated interferon activity, implicating functional effects in dendritic cells and NK cells [175]. Notably, the repertoire of peripherally overexpressed ISGs does not necessarily overlap with genetic risk loci, and these risk effects may be further modulated by epigenetic regulation [75,199]. Recently, among ~200 loci identified by GWAS and exome studies, increasing attention has focused on the contribution of noncoding enhancers/super-enhancers and three-dimensional genome architecture (e.g., CTCF-dependent loops, Z-DNA/RNA) [200].

Inputs that drive interferon production converge on two principal routes: endosomal TLRs and cytoplasmic nucleic acid sensors. Induction of IFN-α by immune complexes—including autoantibodies to RNA-binding proteins (e.g., Ro/Sm), nucleic acids, and products of necrotic or apoptotic cells—correlates strongly with the interferon signature [201,202,203]. Genetic data implicating IRF5 (downstream of TLR7) together with pharmacologic inhibition of TLR7 reinforce the view that TLR7 access to RNA-containing immune complexes is a primary driver [204]. Additional candidate inputs for endosomal TLR activation include neutrophil extracellular traps (NETs), DNA-bearing microparticles, and circulating mitochondrial DNA [205,206,207,208]. Conversely, insights from interferonopathies show that mutations affecting RIG-I/MDA5, the cGAS–STING pathway, and endogenous nucleic-acid–regulatory factors can generate IFN-I signatures. In SLE, oxidized mitochondrial DNA and DNA/RNA derived from endogenous retroelements are plausible inducers of STING-dependent interferon production [209,210,211]. The relative contributions of endosomal TLRs versus cytoplasmic sensors in sporadic SLE—and their roles in monogenic variants—remain to be elucidated.

The tissue niche also modulates interferon outputs. Polymorphisms in IFNK (IFN-κ) have been implicated in cutaneous lupus erythematosus and exhibit sex-dependent effects [212]. Although IFN-κ is not the principal source of circulating IFN-I [174,212], it can amplify local cutaneous inflammation by promoting pDC-dependent IFN production and keratinocyte IL-6 overproduction—processes enhanced by TLR stimulation or UVB exposure in an IFN-κ–dependent manner [213].

### 4.3. Cellular Circuits Executing the IFN Program

#### 4.3.1. DCs

pDCs are regarded as the principal source of IFN-I in SLE, with their output finely tuned by signals from diverse immune cells [214]. In SLE-prone mice, IFN-I–treated DCs exhibit relative resistance to apoptosis, implicating prolonged survival of activated DCs in disease initiation [215]. In lupus-prone mice, pDC depletion mitigates disease severity, and in humans, administration of an anti-BDCA2 antibody—selectively inhibiting pDC function—significantly attenuates the peripheral IFN-I signature [216,217,218]. In SLE patients with cutaneous involvement, anti-BDCA2 therapy markedly reduces IFN-α and MxA within lesional skin, changes that correlate with improved clinical scores and implicate pDC dominance within the cutaneous niche [218]. Nevertheless, the interferon milieu in SLE is not exclusively pDC-dependent: monocytes and conventional DCs (cDCs) also contribute, predominantly via IFN-β production, thereby reinforcing the IFN network through interactions among multiple immune cell types [219,220].

As upstream inputs to pDC activation, exosome-delivered microRNAs have emerged as potent activators, directly stimulating pDCs to augment IFN-I secretion [221]. Moreover, the late endosomal transport inhibitor EGA reduces IFN-α production and release in TLR7-activated pDCs, concurrently decreasing pro-TNF–expressing pDCs and suppressing TNF-α production and release in the R837 stimulation system [222]. In addition, integrin αvβ3, acting as a receptor for apoptotic cell debris, modulates TLR signaling—suppressing autoreactive B-cell activation in lupus models while providing essential contextual cues to pDCs that constrain excessive responses to self nucleic acids [223]. Thus, pharmacological control of the pDC–IFN-I axis constitutes a significant therapeutic objective in SLE [224].

#### 4.3.2. T Cells

T cells undergo positive and negative selection in the thymus and exit to the periphery as naïve cells with defined epitope specificity [225]. Following antigen encounter, in parallel with effector differentiation, a subset acquires regulatory T-cell (Treg) or memory T-cell fates [225]. In SLE, single-cell RNA-seq and clinical studies demonstrate an inverse correlation between IFN-I activity and peripheral lymphocyte or naïve T-cell abundance [226,227]. Although Th1–Th2 disequilibrium is observed in SLE, it is largely contextualized by IFN-γ biology, with comparatively limited delineation of IFN-I-specific effects [228]. In inflamed tissues, increased Th17 and decreased Treg frequencies are reported [229]. IFN-I upregulates Radical S-adenosyl methionine domain-containing protein 2 (RSAD2) in naïve T cells, thereby promoting Th17 and follicular helper T-cell (Tfh) differentiation and contributing to inflammatory cytokine production and disease initiation [230].

Tfh cells—characterized by CXCR5, PD-1, BCL6, BTLA, ICOS, IL-21, and SH2D1A expression—correlate with SLE activity and organ damage and drive autoantibody production through aberrant germinal center (GC) responses [231]. IFN-I enhance STAT4 phosphorylation in Tfh cells, inducing dysregulated IL-21/IFN-γ production [232]. Elevated IFN-I further hyperactivates mTOR, contributing to lymphopenia [233]. In addition, IFN-α upregulates CD25 on Tfh cells, augmenting IL-2–STAT5 signaling; STAT5 then suppresses H3K4me3 at the BCL6 locus and competes with STAT1, reducing BCL6 expression, shifting PD-1^+^CXCR5^+^ Tfh-like cells toward PD-1^+^CXCR5^−^ Tph-like cells and destabilizing the Tfh phenotype [234]. IFN-I also shields Tfh cells from NK cell-mediated cytotoxicity [235].

T peripheral helper (Tph) cells, originally defined in RA, localize to tertiary lymphoid structures and contribute to SLE pathology [236]. Their frequencies increase in active disease and lupus nephritis (LN) [237], and IFN-α-induced Tph-like cells robustly drive B cells toward CD38hiCD27hi plasma cells [234]. Blockade of IFN-α may suppress Tph-like differentiation and plasmacytogenesis [238].

Conversely, SLE is characterized by Treg dysfunction and/or depletion [239]. Although IFN-β can directly promote Treg induction through STAT1/P300-dependent Foxp3 acetylation, IFN-γ-centered pathways remain predominant in the SLE literature [240,241,242].

In LN, CD8^+^ T cells accumulate in periglomerular regions, and their density correlates with disease severity. IFN-I promote CD8^+^ T-cell differentiation, cytotoxicity, and chemotaxis through upregulation of ISGs [243]. In IFNAR-deficient mice, effector CD8^+^ T-cell accumulation is diminished and Treg frequencies increase [244]. Transcriptomically, SLE CD8^+^ T cells differ from healthy controls by aberrations in interferon-stimulated and mitochondrial pathways, and IFN-α has been shown to enhance apoptosis in effector memory CD8^+^ T cells [245].

#### 4.3.3. B Cells

IFN-I upregulate TLR7 on naïve B cells, heightening sensitivity to RNA-associated antigens, while increasing costimulatory molecule expression to augment antigen presentation, T-cell help acquisition, and participation in GC reactions [246,247]. B cells from SLE patients display a pronounced IFN-I–related transcriptional profile, and human B cells can express IFN-I genes in response to IFN-λ stimulation [248].

During this susceptibility phase, IFN-I directly promotes differentiation into CD138^+^ plasmablasts/plasma cells and antibody production in a BCR- and CD40-dependent manner [246]. In vivo, IFN-I fosters B-cell-to-plasmablast transitions, thereby amplifying inflammation and tissue injury [249,250]. In SLE, not only is the mature B-cell pool expanded, but antigen-presenting cells (APC)-like B cells are also enriched within the mature naïve compartment, increasing overall antigen-presenting capacity [251]. Elevated IFN-α augments sensitivity to IL-21 and endosomal TLR ligands, facilitating the emergence of autoreactive age-associated/atypical B cells (ABCs) [252]. IFN-α further enhances BCR signaling, activation, and differentiation, and can partially substitute for B-cell activating factor (BAFF) during the T1-to-T2 transition [252]. Moreover, IFN-I elevate serum BAFF, promoting B-cell proliferation and erosion of immune tolerance [253]. With respect to memory fates, IL-4R and IFN-I receptor signaling regulate the developmental trajectory from transitional B cells to DN1 and classical memory (cMEM) B cells [254].

Moreover, IFN-I expand antibody-forming cells (AFCs) and ICOS^hi^ ExFO^-^ T helper cells, which are critical for the emergence of autoantibody responses [246]. In a subset of SLE patients, the plasmablast-to-memory (PB/M) ratio is markedly elevated and correlates with disease activity (SLEDAI). In this high PB/M group, CD21^low activated switched memory B cells are increased and pre-plasmablasts accumulate. Single-cell transcriptomics of PBs reveals strong expression of proliferation-associated genes and interferon-stimulated genes (e.g., IFI6, IFITM1), indicating activation of IFN receptor pathways centered on IRF7/ISRE and supporting interferon-dependent plasmablast differentiation [255]. Repertoire analyses further show IgG1 dominance with polyclonal expansion, and somatic hypermutation levels comparable to those of memory B cells; serologically, this associates with increased IgG, decreased IgM (a high IgG/IgM ratio), and persistent hypergammaglobulinemia [255]. Notably, anti-Sm/RNP positivity is strikingly frequent, and given their interferon inducibility, a positive feedback circuit linking IFN, plasmablast differentiation, and anti-Sm/RNP production is inferred [255]. Conversely, observations that the presence of anti-IFN-α autoantibodies associates with reduced IFN-I activity and improved clinical indices suggest that long-lived plasma cells sustain their production and that targeted suppression of the IFN axis may ameliorate clinical parameters [256].

This B-cell circuit is also intertwined with metabolic and stress-response pathways. IFN-α enhances mitochondrial function in human CD19^+^ B cells while promoting plasma cell–like differentiation and antibody production [257]. In SLE B cells, the DNA damage response (DDR) is excessively activated, with IFN-I inducing the ataxia telangiectasia mutated and Rad3 related Checkpoint kinase 1 (ATR–Chk1) pathway [258]. Because ATR inhibition suppresses B-cell activation, plasmacytoma formation, antibody production, and inflammatory responses, the metabolic/DDR node of the IFN-dependent circuit represents a viable therapeutic target [258].

Finally, pathogenic B cells contribute to disease through autoantibody production, inflammatory cytokine secretion, and self-antigen presentation [259]. The resulting autoantibodies re-stimulate pDCs as immune complexes (ICs); notably, IgE–ICs amplify IFN-I production by pDCs and exacerbate disease activity [260].

#### 4.3.4. NK Cells, Neutrophils, Monocytes and Macrophages

The innate immune circuitry in SLE is driven by IFN-I and comprises a multilayered feedback system that amplifies these signals. Within the NK cell–DC axis, NK cells have been shown to cooperate with dendritic cells to induce IFN-α production [261].

Neutrophils are a principal source of endogenous antigenic triggers in SLE [262], exhibiting heightened sensitivity to IFN-I and a strong propensity for NET formation [263]. NETs potentiate autoantibody responses by promoting pDC activation and IFN-I secretion [264,265], while the transfer RNA-derived fragment tRF-His-GTG-1 augments IFN-α via TLR8–IRF7 activation [266]. At the clinical genomics level, transcriptomic markers associated with IFN-α/ω inhibition (e.g., JNJ-55920839) have been detected in whole blood, underscoring the pharmacologic plasticity of interferon-pathway targets [267]. Phenotypically, low-density granulocytes (LDGs) display elevated IFN-I production and inflammatory cytokine output and are linked to vascular dysfunction [268]. Conventional neutrophils likewise exhibit increased production of a proliferation-inducing ligand (APRIL), IL-21, and CXCL10 [269]. Collectively, neutrophils contribute to SLE through a triad of antigen provision, interferon amplification, and end-organ injury [270].

Macrophages (Mφ) exhibit M1/M2 plasticity, and in SLE, M2 polarization is diminished, with IRF4 shaping macrophage fate decisions [271,272]. In TLR7-stimulated bone marrow-derived Mφ, aconitate decarboxylase 1 (ACOD1) is induced via IFNAR signaling, promoting immunoregulation and vasoprotection [273]. Conversely, the MyD88-like adaptor protein Mal mediates TLR9-dependent IFN-β/TNF-α transcription through ERK1/2 activation in HSV-1-infected Mφ, engaging noncanonical NF-κB [274]. Systems-level transcriptomics has identified macrophage activation syndrome (MAS) clusters involving IFN-I [275,276]. Metabolically, in SLE patient peripheral blood mononuclear cells (PBMCs), spermine directly binds JAKs, suppressing JAK1 phosphorylation and coordinately restraining IFN-I/II and IL-2/IL-6 pathways [277]. In the kidney, LN macrophages transition dynamically from an inflammatory patrolling state to a phagocytic phenotype and subsequently to an antigen-presenting phenotype [278]. Conversely, plasma 7α,25-dihydroxycholesterol is elevated in SLE and binds Epstein–Barr virus-induced gene 2 (EBI2) to suppress IFN-I responses in Mφ [279]. Collectively, these findings position macrophages as regulatory nodes spanning both the accelerating and braking phases of interferon signaling.

Monocytes comprise chemically distinct “classical” Ly6Chi (CM) and “non-classical” Ly6Clo (NCM) subsets, with CMs identified as the principal source of IFN-I in pristane-induced lupus [280,281]. In SLE, NCMs exhibit defects in DNA repair, heightened cell-cycle and interferon signaling, and skewing toward an M1-like phenotype [282]. SLE monocytes also display a senescent program marked by increased cyclin-dependent kinase inhibitor 2A (CDKN2A), which induces GATA-binding protein 4 (GATA4) and augments IFN-α production via activation of the cGAS–STING pathway, thereby linking cellular aging to inflammation [283]. Clinically, IL-10 and IFN-γ drive CD64 upregulation on monocytes, which correlates with SLEDAI, BUN, and anti-Sm antibodies [284]. Collectively, these findings position monocytes as amplifiers of the IFN-I axis while simultaneously embodying pathological features associated with senescence, defective DNA repair, and altered metabolism.

## 5. Type I Interferon in Rheumatoid Arthritis

### 5.1. Interferon Signatures in Rheumatoid Arthritis: Biomarkers of Therapeutic Response and Subtype Bias

RA is a chronic autoimmune disorder characterized by pain, stiffness, and deformity driven by erosion of articular cartilage and bone [285]. In RA, IFN-I–related markers have emerged as candidate predictors of responsiveness to biologic agents. Pre-treatment interferon gene signature (IGS) positivity in peripheral blood correlates with favorable responses to B-cell depletion therapy with rituximab [286]. By contrast, elevated IFN-I signatures are associated with non-response to anti–TNF therapy [287]. Moreover, the IFN-β–to–IFN-α activity ratio (IFN-β: IFN-α) inversely correlates with responsiveness to anti–TNF therapy; in an independent cohort, a high IFN-β: IFN-α ratio strongly predicted nonresponse to TNF inhibitors [288,289]. Although the biological basis for circulating subtype ratios remains unresolved, cross-disease comparisons indicate IFN-α predominance in SLE and relative IFN-β predominance in RA, suggesting that IFN-β bias in RA may shape therapeutic responsiveness [174,289,290]. Functionally, while early clinical studies posited anti-inflammatory effects of IFN-β, trials of recombinant IFN-β were unsuccessful [291,292,293,294]. This apparent inconsistency with the observation of elevated circulating IFN-β in TNF inhibitor nonresponders likely reflects the substantial plasticity of IFN-I signaling, which is contingent on dose, duration, anatomical site of production, and the surrounding inflammatory milieu [77,289].

### 5.2. Determinants of IFN-I Activity in RA—IGS Dynamics, Susceptibility Loci, Epigenetic Remodeling, and Nucleic-Acid Triggers

Expression of ISG in the synovium was first documented nearly four decades ago, and multiple studies—including recent RNA-seq analyses—have reaffirmed robust IGS expression in synovial tissue [9,295,296,297] (Figure 4) (Box 1). IGS is detectable in peripheral blood and are present even in the preclinical phase [298,299]. Although circulating ISG levels are generally lower than in conditions such as SLE, several interferon-pathway genes are linked to RA risk and may aid in disease prediction [298,300,301,302]. Stage dependence is also evident: elevated IGS (e.g., MxA, OAS1, ISG15, IFI44L, IFI6) are more frequent in early RA and decline with treatment initiation, whereas in established RA the association between IGS and disease activity is inconsistent [303,304,305]. Ontology and network analyses further identify IGS as discriminating DMARD-naïve early arthritis patients who progress to persistent inflammatory arthritis from those who undergo spontaneous remission [306]. In preclinical cohorts, IGS correlates with anti-citrullinated protein antibodies (ACPA) positivity, which associates with imminent RA onset [307,308,309]. By contrast, in established RA, relationships between IGS/IFN-I signaling and autoantibodies (ACPA, rheumatoid factor [RF]) are conflicting, likely reflecting differences in ISG selection, biospecimen matrices, and disease stage [303,304,305,310,311]. Collectively, the interferon program appears phase-dependent: it exhibits strong predictive and stratification value from the preclinical through early stages, with attenuated disease association once RA is established.

GWAS have identified numerous SNPs associated with RA risk, many clustering in genes within the IFN-I response pathway—including DNA sensors, Toll-like receptors, and JAK–STAT mediators. Although the functional consequences of these variants are only partially understood, risk alleles in IRF5, STAT4, and PTPN22—linked to heightened IFN-I signaling in SLE—are likewise associated with RA [198]. Notably, IRF5 polymorphisms correlate with severe, erosive disease, consonant with observations that higher IGS scores in early RA predict refractory phenotypes [304,312].

Epigenetic remodeling (CpG methylation and chromatin reconfiguration) emerges early in RA and varies by cell subset [313]. In untreated early disease, methylation differences have been associated with initial methotrexate responsiveness and with responses to specific biologics in established RA, highlighting epigenetics as a modifier of clinical course and phenotype [313,314,315,316].

Although the drivers of elevated IGS/IFN-α in RA remain incompletely defined, triggers such as viral infections and microbial DNA/antigen fragments have been repeatedly detected in rheumatoid joints [317,318,319]. Noncoding DNA derived from ancient transposable elements (retroelements) may activate intracellular viral sensors and promote localized IFN-I production [320]. Increased retroelement expression has been reported in RA synovium, and in some patients, viral-like transcriptional profiles and IFN-I signaling correlate with high ACPA titers [188,321].

Cell-free nucleic acids (cfDNA) also merit attention. Mice with defects in DNA clearance develop chronic polyarthritis resembling human RA, and elevated cfDNA levels are reproducibly detected in both peripheral blood and synovial fluid in patients with RA [322,323,324].

### 5.3. Cellular and Tissue Drivers of Type I Interferon Signaling in Rheumatoid Arthritis

Neutrophil NETosis—abundant in the synovium—constitutes a potential trigger of IFN-I. NET-derived DNA forms complexes with LL37, secretory leukocyte protease inhibitor (SLPI), or immunoglobulins to activate pDC TLR7/9, thereby inducing IFN-α production [262,325,326,327]. In RA, associations between NETs and ACPA support a vicious cycle in which TNF-α, IL-17A, and even IFN-α itself promote NETosis [325,328].

Regarding lifestyle and environmental modifiers, physical activity inversely correlates with IFN-I signaling; exercise downregulates TLR and IL-17R pathways, reducing production of inflammatory cytokines, including IFN-I [329].

DCs enhance maturation and antigen-presenting capacity by upregulating human leukocyte antigen (HLA)-DR, CD40, CD80, and CD86 in response to IFN-I [330]. This increase in costimulatory molecules enables presentation of self-antigens to low-affinity autoreactive T cells and may contribute to autoimmunity in predisposed individuals [23]. In RA, pDCs are present within the synovium, and expression of ISGs, IFN-α, and IFN-β has been documented, although it remains uncertain whether pDCs constitute the principal source of IFN-α [331,332,333,334,335]. In established RA, synovial pDCs increase while peripheral pDCs decrease; the remaining, albeit immature, pDCs exhibit enhanced IFN-I-producing capacity [336]. Synovial pDCs can produce robust IFN-α, and persistent arthritis has been induced by intra-articular transfer of IFN-I-producing DCs [337,338]; conversely, in other models, pDC deficiency exacerbates arthritis, and disease activity improves with pDC mobilization/activation via TLR7 agonists [339,340,341]. Thus, pDC function appears dualistic, varying with local microenvironment, disease stage, and subset. Moreover, in untreated early RA, whole-blood IGS correlates negatively with CD14^+^ DC frequency—but not with circulating CD1c^+^ DCs or pDCs—highlighting the likelihood that IFN-I signaling drivers differ among DC subsets [340].

Synovial fibroblasts (FLS) are resident stromal cells of the joint that acquire an activated phenotype characterized by apoptosis resistance, heightened proliferation, and increased production of inflammatory mediators [342,343]. In early RA, IFN-α concentrations are comparable in serum and synovial fluid, and chromatin architectural alterations at the IFNAR2 locus correlate with poor prognosis [304,344]. Upregulation of ISGs in FLS is driven not only by direct IFN-I signaling but also by TNF-α–induced autocrine IFN-β production via the mTOR pathway, activating the IRF1–IFN-β–IFNAR–JAK–STAT1 axis; this may underlie the high IRG expression observed in pathogenic sublining FLS (THY1^+^HLA-DRhigh) [297]. Functionally, IFN-α correlates positively with TLR3/7 expression in the synovial lining and sublining and augments downstream IL-6 and TNF production. Moreover, IFN-α markedly enhances TLR4-dependent IL-1β/IL-18 production in synovial cells, whereas IFN-β exerts anti-inflammatory effects by suppressing IL-1β and TNF in PBMCs while dose-dependently increasing IL-1 receptor antagonist (IL-1Ra) [291]. IFN-β also elevates IL-1Ra secretion in FLS and amplifies IL-1β–induced IL-1Ra production in both FLS and chondrocytes [292]. Consistent with these preclinical data, IFN-β administration ameliorated arthritis in collagen-induced models [293,345]; however, in a multicenter, randomized, double-blind phase II trial, subcutaneous recombinant IFN-β failed to demonstrate efficacy in active RA [294].

Neutrophils play a pivotal role in RA pathogenesis and associate with heightened clinical disease activity and exacerbated inflammatory status [346]. Polymorphonuclear granulocytes (PMNs) exhibit higher IFNAR expression than PBMCs from either healthy controls or RA patients and constitute major contributors to the whole-blood IGS in RA [347]. Next-generation sequencing of circulating neutrophils from RA patients demonstrates significantly increased IRG expression relative to healthy controls [348]. Nonetheless, the peripheral interferon signature in RA is generally weaker than in SLE, necessitating attention to disease-specific differences in signal intensity [349].

Classical and nonclassical monocytes contribute to RA pathogenesis [350]. Although the in vivo impact of the IGS on monocyte function remains uncertain, in vitro exposure to IFN-I upregulates TLR7 and IRF expression in monocytes, heightening their responsiveness to subsequent immunostimulatory ligands [351]. In parallel, CD40, CD80, CD86, and HLA-DR are increased, promoting differentiation into highly antigen-presenting monocyte-derived dendritic cells (moDCs), which are expanded in RA synovium and can drive Th17 differentiation [330,352]. These effects are subset dependent; for instance, in murine models, inflammatory monocytes may secondarily augment IFN-α responsiveness via increased IFNAR expression, indicating that behavior varies with the in vivo context [219].

IFN-I broadly augments B-cell activity by inducing BAFF production from monocytes, directly stimulating B cells, and indirectly supporting B-cell survival via T-cell and DC activation [353]. Consequently, IFN-I promote long-term B-cell survival, facilitates differentiation into memory and plasma cells, drives isotype switching, and enhances autoantibody formation, while shifting the plasma-cell transcriptome toward a proinflammatory state [354,355]. Mechanistically, IFNAR-mediated modulation of BCR signaling can potentiate pathways underpinning antibody formation and germinal center responses, as shown in murine models [250]. In parallel, IFN-I skew CD4^+^ T cells toward Th1 differentiation—thereby amplifying B-cell activation—and enhances CD8^+^ T-cell survival and cytotoxicity, sustaining proliferation and expansion of antigen-specific CD8^+^ T cells by limiting apoptosis [356,357,358]. Within RA synovium, CD8^+^ T cells express IFN-γ more frequently than CD4^+^ T cells, indicating that they constitute the principal intratissue source of IFN-γ and a pathogenic subset capable of driving the local interferon signature [359].

## 6. Type I Interferon in Vasculitis

The contribution of IFN-I to vasculitis varies substantially across disease entities. In a cross-sectional cohort, interferon signatures did not reliably distinguish patients with giant cell arteritis (GCA) or polymyalgia rheumatica (PMR) from controls, and IFN-I–inducible serum markers were uninformative for initial diagnosis or monitoring [360]. By contrast, multiple tissue-level studies demonstrate heightened IFN-I activity in affected vessels: MxA—a prototypic IFN-I-induced protein—is detected in temporal artery biopsies, and aortic transcriptomes show robust induction of type I/II interferon programs [10,14,361]. Network analyses place IFN–JAK–STAT nodes—including STAT3, IRF7, STAT1, and IRF1—at the core, implicating interferon signaling as a primary pathway in arteritis [14]. Experimental models recapitulate increased type I/II interferon transcripts in human temporal artery transplant systems [361]. Clinically, the reduction in ISG expression following prednisolone therapy suggests that interferon responses are already active at disease onset [362]. In some series, pDCs are not detected on temporal artery biopsy (TAB), implicating non-pDC sources—such as macrophages—as producers of IFN-I [10,363].

In GCA, IL-6 and IFN-γ are major disease drivers, both signaling via JAK–STAT [364]; the incomplete suppression of inflammation by IFN-γ blockade alone suggests a contributory role for IFN-I [364]. Corroborating this, baricitinib has shown efficacy in relapsing GCA, and upadacitinib demonstrated efficacy and subsequently received FDA approval for GCA (April 2025) [20,365,366].

Reports implicating IFN-I in vasculitis other than GCA are limited. Although IFN-I responses are not considered primary drivers in human ANCA-associated vasculitis, a murine model shows that cGAS–STING activation worsens disease, with IFN-β produced by monocyte-derived macrophages acting as a decisive effector [367,368]. In Takayasu arteritis (TAK), current evidence likewise suggests that IFN-I are not a major pathogenic driver [369].

## 7. Therapeutic Implications and Stratification Strategies

Therapeutics targeting the IFN-I axis can be organized into three domains: inhibition of IFN-I production, neutralization of IFN-I itself, and blockade of downstream signaling (Table 2).

### 7.1. Upstream Strategies to Limit Type I Interferon Production: Receptor Blockade, pDC Targeting, and Sensor/Transcriptional Modulation

Strategies to curb IFN-I production proceed sequentially from receptor-level interception to depletion of producing cells and inhibition of intracellular sensors/transcriptional programs. Classically, hydroxychloroquine (HCQ) impedes endosomal acidification and TLR7/9 signaling, reducing IFN-α and TNF production by pDCs upon TLR7/9 stimulation in SLE [370]. Clinically, HCQ is recommended for all patients with SLE—where not contraindicated—to reduce flare frequency and severity and prevent disability, and it is widely employed as part of combination therapy in RA [371,372].

Direct pDC targeting includes the anti-BDCA2 antibody litifilimab, which suppresses pDCs and diminishes IFN production in cutaneous lupus lesions, and anti-CD123 antibodies that similarly attenuate IFN-I via pDC engagement [224,373]. Along the intracellular sensor axis, cordycepin may temper IFN-I excess by promoting autophagy-mediated degradation of STING after DNA stimulation [374]. The CXCR4 agonist clobenpropit robustly suppresses IRF7 phosphorylation, lowers IFN production, and reduces inflammatory cytokines in a lupus model [375]. At the transcriptional tier, HDAC10 inhibits IRF3 phosphorylation; in SLE, enhanced autophagic degradation of HDAC10 suggests a mechanistic link with the clinical use of autophagy inhibitors (chloroquine/HCQ) [376]. Upstream receptor blockade includes oral TLR7/8 inhibitors (enpatoran, afimetoran), now in phase II trials for SLE [377,378,379]. Modulation of circulating immune-cell trafficking with the selective S1P receptor modulator cenerimod reduces interferon-related biomarkers [380,381]. Additionally, methylprednisolone pulse therapy promotes Treg differentiation by inducing CD4^+^ T-cell apoptosis and enhancing monocyte TGF-β production, thereby fostering an immunoregulatory milieu that suppresses CD4^+^ proliferation and IFN-γ production [382].

Collectively, a multilayered program—spanning the TLR–pDC–STING–IRF continuum and incorporating HCQ, pDC-directed antibodies, TLR7/8 antagonists, STING degradation, inhibition of IRF phosphorylation, and S1P-pathway modulation—is being implemented. Precise, upstream inhibition tailored to disease- and tissue-specific IFN drivers will be pivotal going forward.

### 7.2. Neutralizing IFN-α and Blocking IFNAR

Neutralization of IFN-α with monoclonal antibodies has shown mixed results: sifalimumab improved disease activity and reduced the interferon signature in SLE, whereas rontalizumab did not meet its primary endpoint but suggested efficacy in patients with low pretreatment interferon signatures [383,384,385]. In addition, IFN-α quinones have been reported to elicit polyclonal anti–IFN-α humoral responses with potential clinical benefit [386]. In idiopathic inflammatory myopathies, sifalimumab similarly reduced the circulating interferon signature [387].

Receptor-level blockade offers broader coverage: anifrolumab (anti-IFNAR1) inhibits signaling downstream of both IFN-α and IFN-β, suppresses ISGs, and confers multidomain clinical improvement in SLE, leading to regulatory approval [19,169,266,388]. In RA, a small pilot study enriched for high-IGS cases showed a preliminary efficacy signal, warranting further validation [389,390]. QX006N, which similarly targets the SD3 domain of IFNAR1 to prevent receptor complex formation, is also in clinical development [391]. Across trials, elevated pretreatment interferon signatures have repeatedly emerged as candidate predictors of therapeutic responsiveness, underscoring the value of biomarker-based stratification across diseases [383,385,392].

### 7.3. Downstream Attenuation of IFN-I Signaling

Downstream of IFNAR, the JAK–STAT pathway constitutes the central signaling node, with TYK2 integrating both IFN-I and IL-12/23 pathways [393]. In SLE, IL-12 stimulation coactivates STAT1 and STAT4, expanding Tfh–Th1–like populations and underscoring the therapeutic appeal of this axis [394]. JAK/TYK2 inhibition attenuates interferon signaling: the JAK1/2-selective inhibitor baricitinib improves cutaneous and articular manifestations, and the JAK1-selective filgotinib likewise ameliorates skin disease [395,396,397]. Upadacitinib has progressed to phase III evaluation, with ongoing confirmation of efficacy [398,399]. TYK2 inhibitors may suppress the interferon signature and the IL-12 axis while preserving IL-2-dependent Treg differentiation [400]; the highly selective inhibitor deucravacitinib potently abrogated IFN-α-induced lymphopenia [401]. The JAK2 inhibitor ruxolitinib also reduces ISG expression and JAK–STAT activity [402]. Mechanistically, JAK blockade limits STAT phosphorylation, thereby diminishing IRG expression, and inhibits IFN-I-dependent plasmacytoid differentiation, synovial BAFF production, CD80/CD86 upregulation on moDCs, and Th1/Th17 differentiation [198,403,404,405]. Notably, secondary analyses of phase II SLE trials with baricitinib indicate that clinical efficacy is independent of reductions in the interferon gene signature [406].

IFN-I induce the B-cell activating factor BAFF; accordingly, belimumab, which disrupts this downstream axis, has established efficacy in SLE and, when added for active lupus nephritis, increases remission rates and reduces relapse [407]. Moreover, pretreatment interferon signature scores have been reported to predict 12-month SRI responses to belimumab [408]. Preclinical data further suggest that belimumab mitigates lupus-like pathology by modulating the V-domain immunoglobulin suppressor of T-cell activation (VISTA) pathway and thereby regulating IFN-I/noncanonical NF-κB signaling [409]. In addition, small molecules targeting the upstream TBK1/IKKε node (selective JAK3/JAK1/TBK1 inhibitors) suppress IFN-I production and osteoclastogenesis in mice, with efficacy demonstrated in autoimmune arthritis models [410,411,412,413].

From a safety standpoint, JAK inhibitors—which attenuate the JAK–STAT pathway integral to Th1 maturation—are associated with increased risk of intracellular infections such as herpes zoster and tuberculosis, both theoretically and in clinical practice [414,415,416]. Pre-initiation vaccination is recommended [414]. More broadly, a two-tiered therapeutic paradigm—comprising IFN receptor blockade and attenuation of downstream signaling—may facilitate more refined stratification by disease subtype, biomarker profile, and infection risk.sectio.

**Table 2 biomolecules-15-01586-t002:** IFN-I Axis Therapies.

Class	Agent	Target	Outcome	Safety	Refs.
**Upstream modulators**	**Hydroxychloroquine (HCQ)**	Endosomal pH increase; TLR7/9 inhibition in pDCs; autophagy/HDAC10 linkage in SLE	First-line in SLE; flares/severity reduced; used in RA combinations	Ocular toxicity	[370,371,372,376,417]
	**Litifilimab; anti-CD123**	pDC targeting; IFN production reduced	IFN signature reduced in cutaneous lupus lesions	Infection risk	[224,373,418]
	**Clobenpropit**	CXCR4 agonism; IRF7 phosphorylation reduced; IFN production lowered	IFN and inflammatory cytokines reduced in lupus model	Preclinical	[375]
	**Enpatoran; Afimetoran**	Selective oral TLR7/8 inhibitors	Rapid suppression of IFN-I gene signature with early clinical signals in CLE/SLE	Phase 2 programs ongoing; safety profile still being defined	[377,378,379]
	**Cenerimod**	Selective S1P receptor modulator	IFN-associated proteins and IFN-1/IFN-γ/plasma-cell signatures reduced; larger effect at 4 mg	Dose-related lymphopenia	[380,381]
**IFN-I–directed agents**	**Sifalimumab; Rontalizumab**	Anti–IFN-α mAbs	Sifalimumab: disease activity and IFN signature reduced; Rontalizumab: primary endpoint not met overall (signal in low IFN-signature subgroup)	Infections	[383,384,385]
	**Anifrolumab**	IFNAR1 blockade; ISGs reduced	Multidomain clinical improvement; regulatory approval in SLE	Herpes zoster and other infections	[19,169,266,388,389,390]
	**QX006N**	IFNAR1 SD3 binding; receptor complex formation prevented	In clinical development	—	[391]
**Downstream modulators**	**Baricitinib**	JAK1/2 inhibition	Improves cutaneous/articular disease; efficacy not strictly tied to IFN-signature reduction	Infections	[395,396]
	**Filgotinib; Upadacitinib**	JAK1 inhibition	Filgotinib: skin disease improvement; Upadacitinib: phase III ongoing	Infections	[397,398,399]
	**Ruxolitinib**	JAK2 inhibition	ISG expression and JAK–STAT activity reduced; signal in IFN-driven states	Infections	[402]
	**Deucravacitinib**	Selective allosteric TYK2 inhibitor	Higher SRI-4 and secondary responses vs. placebo	Infections	[401]
	**Belimumab**	BAFF neutralization	Established efficacy in SLE; added to active LN increases remission and reduces relapse	infections	[407,408,409]

IFN, interferon; IFN-I, type I interferon; IFN-γ, interferon-gamma (type II interferon); IFNAR1, interferon-α/β receptor 1; ISG(s), interferon-stimulated gene(s); JAK, Janus kinase; JAK–STAT, Janus kinase–signal transducer and activator of transcription pathway; TYK2, tyrosine kinase 2; pDC(s), plasmacytoid dendritic cell(s); TLR7/9, Toll-like receptor 7/9; CXCR4, C-X-C chemokine receptor 4; IRF7, interferon regulatory factor 7; S1P/S1P1, sphingosine-1-phosphate/S1P receptor 1; BAFF, B-cell activating factor; HCQ, hydroxychloroquine; HDAC10, histone deacetylase 10; mAb, monoclonal antibody; SLE, systemic lupus erythematosus; CLE, cutaneous lupus erythematosus; RA, rheumatoid arthritis; LN, lupus nephritis.

## 8. Conclusions

IFN-I occupy a pivotal position at the interface of antiviral defense and breakdown of self-tolerance, with effects that appear highly context dependent across diseases, tissues, and disease stages. Convergent nucleic-acid sensing and IFNAR–JAK–STAT signaling, modulated by post-transcriptional control, epigenetic modifications, and non-coding RNAs, yield interferon-stimulated gene signatures that are typically more pronounced in systemic lupus erythematosus and more heterogeneous across rheumatoid arthritis and vasculitis. Therapeutic strategies that target the IFN-I axis may mitigate aberrant inflammation; however, broader implementation is constrained by incomplete mechanistic delineation, limited high-quality preclinical and clinical evidence, and the challenge of selectively modulating inflammation without compromising host defense. Further work is warranted to map IFN-I regulatory networks across disease trajectories, refine and validate biomarker panels for patient stratification and monitoring, and design trial frameworks aligned to mechanistic endotypes.

## Figures and Tables

**Figure 1 biomolecules-15-01586-f001:**
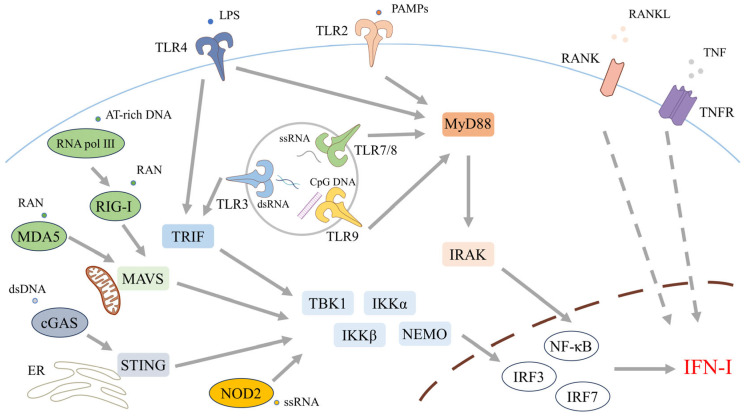
Pattern-recognition receptor (PRR) pathways that initiate type I interferon (IFN-I) production. IFN-I induction is triggered when PRRs sense exogenous or endogenous nucleic acids. At the plasma membrane, Toll-like receptor (TLR)4 recognizes lipopolysaccharide (LPS) from Gram-negative bacteria, while TLR2—typically in heterodimers with TLR1 or TLR6—detects a broad range of pathogen-associated molecular patterns (PAMPs) from bacteria, fungi, parasites, and viruses. Endosomal TLRs chiefly sense nucleic acids: TLR3 detects double-stranded RNA (dsRNA), TLR7 and TLR8 recognize single-stranded RNA (ssRNA), and TLR9 recognizes unmethylated CpG DNA. TLR signaling proceeds via MyD88-dependent and MyD88-independent branches. All TLRs except TLR3 employ MyD88 to activate NF-κB via IRAK family kinases, whereas TLR3 and TLR4 engage TRIF (with TRAM for TLR4) to promote TRAF3-dependent activation of TBK1/IKKε, culminating in IRF3 phosphorylation and dimerization. In the cytosol, the RIG-I-like receptors (RLRs) RIG-I and MDA5 sense viral RNA and signal via the adaptor MAVS to TBK1/IKKε, thereby activating IRF3 and IRF7. In addition, AT-rich DNA can be transcribed by RNA polymerase III into 5′-triphosphate RNA that serves as a RIG-I agonist. The cGAS–STING pathway detects cytosolic double-stranded DNA (dsDNA): cGAS generates cyclic GMP–AMP (cGAMP), which binds STING on the endoplasmic reticulum (ER) to recruit and activate TBK1, driving IRF3 nuclear translocation and IFN-I gene induction. NOD-like receptors NOD1 and NOD2, which sense bacterial and viral signatures, can further contribute to IFN-I responses. Collectively, phosphorylated IRF3/IRF7 together with NF-κB bind IFN-I promoters to initiate transcription of IFN-I and ISGs. Beyond PRRs, members of the tumor necrosis factor receptor superfamily—including RANK—can reinforce ISG expression through paracrine and autocrine signaling.

**Figure 2 biomolecules-15-01586-f002:**
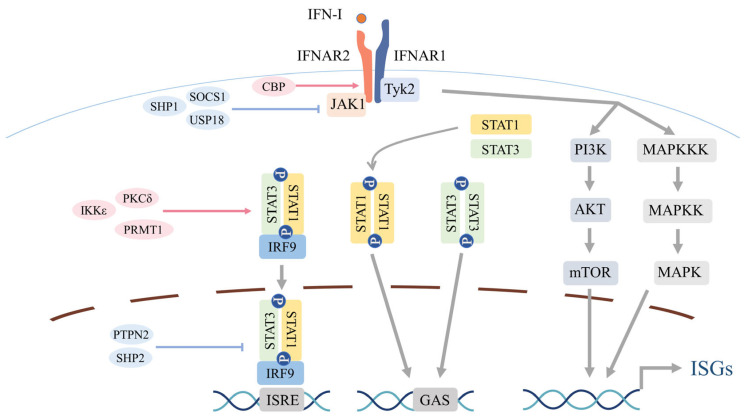
Canonical and noncanonical signaling downstream of the type I interferon (IFN-I) receptor. All nucleated cells express the transmembrane IFN-I receptor, typically a heterodimer of IFNAR1 and IFNAR2. Ligand engagement brings together IFNAR1-associated TYK2 and IFNAR2-associated JAK1, driving receptor rearrangement and activation of the receptor-bound Janus kinases. This leads to phosphorylation of tyrosine residues on IFNAR and subsequent phosphorylation of STAT1 and STAT2. Phosphorylated STAT1 and STAT2 dimerize and, with IRF9, form the interferon-stimulated gene factor 3 (ISGF3) complex, which translocates to the nucleus, binds interferon-stimulated response elements (ISREs), and induces ISGs. Under specific contexts, STAT1 or STAT3 homodimers bind gamma-activated sequence (GAS) to drive distinct ISG modules. In parallel, IFNAR activation can signal through STAT-independent, noncanonical cascades—including phosphoinositide 3-kinase (PI3K)–AKT and mitogen-activated protein kinase (MAPK) pathways—to amplify cellular responses. Multiple regulatory checkpoints tune pathway amplitude and duration. Proximally, SOCS family proteins restrain JAK1/TYK2 and limit STAT phosphorylation, while USP18 associates with IFNAR2 to competitively limit JAK1 recruitment. By contrast, PKCδ-mediated STAT1 Ser727 phosphorylation and IKKε-mediated STAT1 Ser708 phosphorylation enhance transcriptional output, and protein arginine methyltransferase 1 (PRMT1) augments STAT1 DNA binding and transactivation. Within the nucleus, phosphatases such as PTPN2 and SHP2 (PTPN11) dephosphorylate STATs to temper signaling. Chromatin-level control includes context-dependent histone acetylation by CBP and GCN5 at ISG loci. Beyond these mechanisms, IFN-I signaling is further modulated by ubiquitination and ubiquitin-like modifications (e.g., SUMOylation, ISGylation).

**Figure 3 biomolecules-15-01586-f003:**
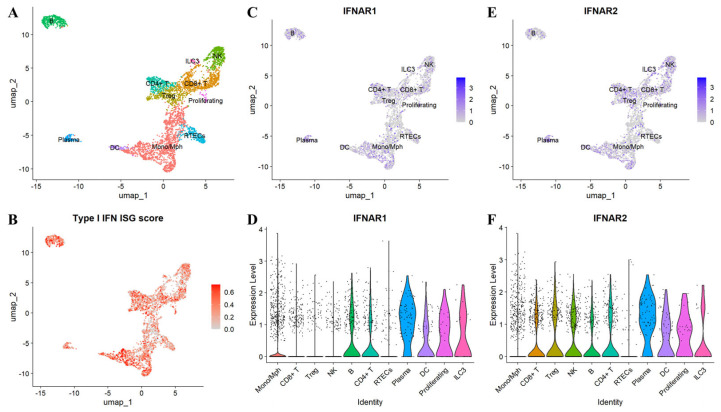
Interferon-stimulated gene (ISG) signature and IFNAR1/2 expression in renal tissue from patients with systemic lupus erythematosus (SLE). Single-cell RNA sequencing data from the Accelerating Medicines Partnership (AMP) RA/SLE Network [190] were re-analyzed to characterize ISG activity and IFNAR1/2 expression in lupus nephritis kidney tissue. (**A**) Annotated UMAP plot showing major renal and immune cell populations. (**B**) UMAP feature plot depicting the aggregated ISG signature across single cells. (**C**) UMAP feature plot of IFNAR1 transcript abundance. (**D**) Violin plot showing IFNAR1 expression by annotated cell type. (**E**) UMAP feature plot of IFNAR2 transcript abundance. (**F**) Violin plot showing IFNAR2 expression by annotated cell type. RTECs, Renal tubular epithelial cells.

**Figure 4 biomolecules-15-01586-f004:**
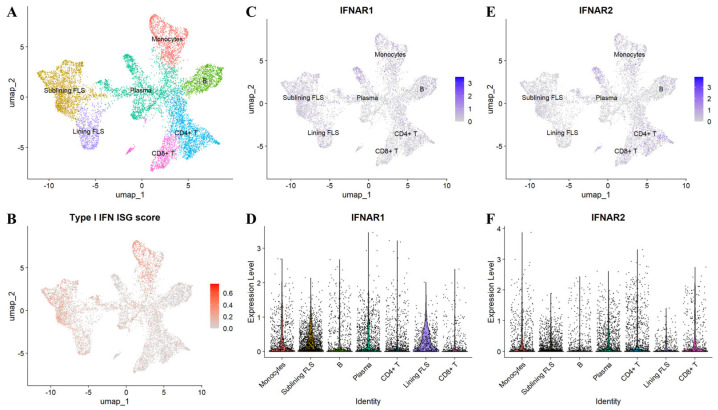
Interferon-stimulated gene (ISG) signature and IFNAR1/2 expression in rheumatoid arthritis (RA) synovial tissue. Single-cell RNA sequencing data from the Accelerating Medicines Partnership (AMP) RA/SLE Network [297] were re-analyzed to profile ISG activity and IFNAR1/2 expression in inflamed synovium from patients with RA. (**A**) UMAP embedding with annotated major synovial cell populations. (**B**) UMAP feature plot of the aggregated ISG score across single cells. (**C**) UMAP feature plot of IFNAR1 transcript abundance. (**D**) Violin plot showing IFNAR1 expression by annotated cell type. (**E**) UMAP feature plot of IFNAR2 transcript abundance. (**F**) Violin plot showing IFNAR2 expression by annotated cell type.

**Table 1 biomolecules-15-01586-t001:** Positive and Negative Regulators of IFN-I Signaling.

Regulator	Level/Target	Mechanism	Net Effect	Refs.
SOCS family	Receptor–proximal; JAK1/TYK2; IFNAR1	Inhibit JAK1/TYK2; block STAT recruitment/phosphorylation via IFNAR1 binding	Negative	[95,96]
USP18	Receptor-proximal (IFNAR2/JAK1); ISGylation axis	Competes with JAK1 at IFNAR2 (receptor brake) and acts as the principal de-ISGylase removing ISG15 from substrates	Negative	[97,98]
SHP1/PTPN6, SHP2/PTPN11, PTPN2	Receptor-proximal and nuclear JAK–STAT	Dephosphorylate receptor/JAK/STAT components—including nuclear STAT1—to terminate signaling and limit the IFN signature	Negative	[99,100]
PKCδ → STAT1(Ser727)	STAT1	Phosphorylates Ser727 to boost STAT1 transcriptional output	Positive	[101]
IKKε → STAT1(Ser708)	STAT1	Phosphorylates Ser708, enhancing DNA binding and ISG transcription	Positive	[102]
PRMT1 → STAT1 (Arg methylation)	STAT1	Arginine methylation enhances STAT1 DNA binding and transactivation, amplifying ISG expression	Positive	[103]
CBP/GCN5	Histones proximal to STAT complexes	Acetylate nearby histones in a context-dependent manner, modulating chromatin accessibility and ISG transcription	Context-dependent	[104,105]
STAT1 hyperacetylation	STAT1	Impedes STAT1 phosphorylation, nuclear translocation, and DNA binding	Negative	[106]
HDAC3	STAT1	Deacetylation counterbalances inhibitory hyperacetylation, restoring STAT1 function	Positive	[106]
Ubiquitination	Pathway-wide (IFNAR1; STAT1/STAT4)	K48 chains drive proteasomal degradation (e.g., IFNAR1 via SCF (HOS); STATs via SLIM/Smurf1) to dampen signaling; K63 chains support signal propagation; NKLAM promotes STAT1 phosphorylation/transactivation	Context-dependent	[107,108,109,110,111,112]
SUMOylation (PIAS1 → STAT1 Lys703)	STAT1	SUMOylation suppresses STAT1 activity and reduces ISG expression	Negative	[113]
ISGylation (ISG15 via UBE1L–UBCH8–HERC5)	Pathway-wide substrates	Covalent ISG15 conjugation that generally potentiates IFN-I signaling	Positive	[98]
Histone acetylation axis (BRD4–P-TEFb; HDACs incl. HDAC1–PLZF)	ISG chromatin/transcriptional elongation	Acetylation recruits BRD4–P-TEFb to promote elongation; HDAC activity remodels/limits ISG programs (HDAC1 recruits PLZF)	Context-dependent	[105,114,115,116]
H3K9me2	ISG chromatin	Repressive histone mark that limits ISG induction	Negative	[117,118]
H2B monoubiquitination	ISG chromatin	IFN-induced H2B-ub promotes chromatin opening; PARP9–DTX3L supports H2B-ub and ISG transcription	Positive	[119,120]
DNA methylation/TET–5hmC axis	ISG chromatin/DNA	DNMT3A/3B install 5mC (generally repressive); TET-mediated 5hmC participates in transcriptional control; ISG-specific roles remain unresolved	Context-dependent	[121,122,123,124]
microRNAs (miRNAs)	IFNAR, JAK–STAT components, ISG mRNAs	Post-transcriptional silencing that can dampen or enhance IFN-I signaling (e.g., miR-29a ↓IFNAR1; miR-155 ↓SOCS1 → ↑signaling)	Context-dependent	[125,126,127]
long noncoding RNAs (lncRNAs)	Chromatin/RNA regulatory layer	Scaffold and cis-regulatory functions that tune IFN output (e.g., lnc-DC → STAT3 phosphorylation ↑; IFN-inducible lncRNAs; NeST/Tmevpg1 → H3K4me3 via WDR5 ↑; NRAV ↓ ISG program)	Context-dependent	[128,129,130,131]

IFN, interferon; IFN-I, type I interferon; IFNAR, interferon-α/β receptor; ISG(s), interferon-stimulated gene(s); JAK, Janus kinase; TYK2, tyrosine kinase 2; STAT, signal transducer and activator of transcription; SOCS, suppressor of cytokine signaling; USP18, ubiquitin-specific protease 18; SHP1/PTPN6, Src homology region 2 domain-containing phosphatase-1/protein tyrosine phosphatase non-receptor type 6; SHP2/PTPN11, protein tyrosine phosphatase non-receptor type 11; PTPN2, protein tyrosine phosphatase non-receptor type 2; PKCδ, protein kinase C delta; IKKε, IκB kinase epsilon; PRMT1, protein arginine methyltransferase 1; PIAS1, protein inhibitor of activated STAT1; SLIM, STAT-interacting LIM protein; Smurf1, SMAD-specific E3 ubiquitin protein ligase 1; NKLAM, natural killer lytic-associated molecule; ISG15, interferon-stimulated gene 15; HAT, histone acetyltransferase; CBP, CREB-binding protein; GCN5, general control nonderepressible 5; HDAC, histone deacetylase; BRD4, bromodomain-containing protein 4; P-TEFb, positive transcription elongation factor b; PLZF, promyelocytic leukemia zinc finger; H3K9me2, histone H3 lysine-9 dimethylation; H2B-ub, histone H2B monoubiquitination; RNF20/hBre1, ring finger protein 20/human Bre1; PARP9–DTX3L, poly(ADP-ribose) polymerase 9–deltex 3-like complex; DNMT3A/3B, DNA methyltransferase 3A/3B; TET, Ten-eleven translocation dioxygenases; 5mC, 5-methylcytosine; 5hmC, 5-hydroxymethylcytosine; miRNA, microRNA; lncRNA, long noncoding RNA; RBP, RNA-binding protein.

## Data Availability

The AMP RA/SLE Network data used for this publication are available at https://arkportal.synapse.org/Explore/Programs/DetailsPage?Program=AMP-RA/SLE (accessed on 30 September 2025).

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
