# Peer review of "The Type I Interferon Axis in Systemic Autoimmune Diseases: From Molecular Pathways to Targeted Therapy"

_biomolecules, 2025, doi:10.3390/biom15111586_

Round 1
Reviewer 1 Report
Comments and Suggestions for Authors
General Overview
This is an ambitious and impressively comprehensive review that synthesizes the molecular and cellular pathways of Type I interferon (IFN-I) biology with their implications in systemic autoimmune diseases, notably systemic lupus erythematosus (SLE), rheumatoid arthritis (RA), and vasculitis. The paper demonstrates clear command of the literature and provides a logically organized, mechanistic narrative that moves from signaling pathways to disease relevance and therapeutic targeting. The manuscript is scholarly and up to date, reflecting the authors’ expertise in clinical immunology and molecular mechanisms.
However, the review would benefit from greater historical contextualization, improved integration of early foundational work on cytokine-mediated autoimmunity (particularly that of Skurkovich), and some refinements in focus, readability, and citation structure.
Strengths
- Comprehensive coverage – The review is exhaustive in scope, detailing the upstream sensors (TLRs, RLRs, cGAS-STING), canonical and noncanonical IFNAR signaling, and multi-layered regulation, extending through cellular and epigenetic mechanisms.
- Scientific accuracy and currency – The discussion of IFN-I–associated biomarkers, genetic loci, and JAK-STAT-targeted therapeutics is well supported by recent transcriptomic and clinical data, consistent with contemporary understanding (e.g., AMP RA/SLE datasets, anti-IFNAR therapy, JAK inhibitors).
- Mechanistic clarity – The authors effectively link molecular mechanisms to disease phenotypes, helping bridge basic science to translational immunology.
- Illustrative figures – Figures appear to be conceptually clear and consistent with the text, facilitating comprehension of complex signaling networks.
- Logical organization – The progression from IFN biology to specific autoimmune diseases and then to therapeutic strategies is coherent and pedagogically sound.
Weaknesses and Areas for Improvement
- Lack of historical context
The manuscript’s review of the pathogenic role of pro-inflammatory cytokines omits early, foundational contributions. Long before the identification of IFN signatures or JAK inhibitors, Simon Skurkovich proposed the “cytokine theory of autoimmune disease,” emphasizing interferons and related pro-inflammatory mediators as central pathogenic factors. Their pioneering work in the 1970s–1990s established many of the conceptual underpinnings later confirmed by molecular genetics.
Recommendation:
Include historical references to:
- Skurkovich SV, Klinova EG, Eremkina EI, Levina NV. Immunosuppressive effect of an anti-interferon serum. Nature. 1974 Feb 22;247(5442):551-2.
- Skurkovich S, Skurkovich B, Bellanti JA. A unifying model of the immunoregulatory role of the interferon system: can interferon produce disease in humans? Clin Immunol Immunopathol. 1987 Jun;43(3):362-73.
- Skurkovich S, Skurkovich B. Anticytokine therapy, especially anti-interferon-gamma, as a pathogenetic treatment in TH-1 autoimmune diseases. Ann NY Acad Sci. 2005 Jun;1051:684-700.
These citations would acknowledge the evolution of the field and strengthen the review’s scholarly depth.
- Excessive mechanistic density
While mechanistic detail is commendable, several sections (e.g., 2.3–2.5) are encyclopedic and could be condensed or summarized in schematic tables to improve readability. The repetition of signaling intermediates (e.g., JAK/STAT modifiers, histone modifiers) detracts from the overall flow.
Recommendation:
Condense the description of canonical IFNAR–JAK–STAT signaling and summarize noncanonical and regulatory pathways in tabular format (e.g., “Positive and Negative Regulators of IFN-I Signaling”).
- Limited discussion of clinical translation
The section on “Targeted Therapy” is brief relative to the extensive mechanistic background. Readers would benefit from a concise table summarizing:
- IFN-I–targeted therapies (anifrolumab, sifalimumab, rontalizumab, etc.)
- JAK inhibitors (baricitinib, tofacitinib, upadacitinib)
- Associated clinical outcomes and adverse effects.
- Integration of cross-disease insights
The review addresses SLE, RA, and vasculitis separately but does not fully develop their shared IFN-I signatures and potential for comparative endotyping. A synthesis section comparing these conditions (perhaps titled “Common and Divergent IFN-I Mechanisms Across Autoimmune Diseases”) would strengthen the paper’s translational impact.
- Citation and referencing
While most citations are recent, the paper occasionally over-relies on secondary reviews. Incorporating primary studies — especially those identifying IFN-I signatures and therapeutic targets — would increase rigor. Additionally, citations should be checked for completeness (journal, year, volume).
- Stylistic and editorial considerations
- Some sentences are excessively long and could be split for readability.
- The abstract could be made more concise by removing redundant phrases such as “context-dependent, biphasic IFN response governed by dose, timing, and tissue microenvironment.”
- Minor grammatical edits are needed (e.g., consistent use of “IFN-I,” “IFN-Is,” or “Type I IFNs”).
Specific Suggestions
|
Section |
Recommendation |
|
Introduction |
Add a short historical paragraph crediting early recognition of interferons and pro-inflammatory cytokines as mediators of autoimmunity — referencing Skurkovich. |
|
Signaling Pathways (2.3–2.5) |
Consolidate into a more synthetic format, possibly with a schematic figure or table of key regulators. |
|
Epigenetic Regulation (2.5) |
Discuss in relation to emerging therapeutic epigenetic modifiers. |
|
SLE Section (4) |
Include a concise model figure summarizing the feedback loop between pDCs, IFN-I, and autoantibody production. |
|
Therapeutic Implications |
Expand discussion of current and investigational IFN-targeted treatments with clinical trial data. |
|
Conclusion |
End with forward-looking statements on precision medicine and biomarker-guided therapy. |
Conclusion
This manuscript is scientifically sound, exhaustive, and potentially publishable after revision. Its main improvement opportunity lies in balancing depth with readability and acknowledging historical perspectives.
Incorporating references to the pioneering work of Simon Skurkovich and colleagues who were among the first to articulate the pathogenic role of interferons and pro-inflammatory cytokines in autoimmune disease, would provide valuable historical continuity and intellectual depth.
Author Response
Comments to be transmitted to the Author
1. Lack of historical context
The manuscript’s review of the pathogenic role of pro-inflammatory cytokines omits early, foundational contributions. Long before the identification of IFN signatures or JAK inhibitors, Simon Skurkovich proposed the “cytokine theory of autoimmune disease,” emphasizing interferons and related pro-inflammatory mediators as central pathogenic factors. Their pioneering work in the 1970s–1990s established many of the conceptual underpinnings later confirmed by molecular genetics.
Recommendation:
Include historical references to:
1.Skurkovich SV, Klinova EG, Eremkina EI, Levina NV. Immunosuppressive effect of an anti-interferon serum. Nature. 1974 Feb 22;247(5442):551-2.
2.Skurkovich S, Skurkovich B, Bellanti JA. A unifying model of the immunoregulatory role of the interferon system: can interferon produce disease in humans? Clin Immunol Immunopathol. 1987 Jun;43(3):362-73.
3.Skurkovich S, Skurkovich B. Anticytokine therapy, especially anti-interferon-gamma, as a pathogenetic treatment in TH-1 autoimmune diseases. Ann NY Acad Sci. 2005 Jun;1051:684-700.
Response: Thank you for highlighting these foundational contributions. We have added a brief historical background tracing the field from the discovery of interferons by Isaacs and Lindenmann to the conceptual framework proposed by Skurkovich and colleagues, emphasizing their early articulation of the “cytokine theory of autoimmune disease” and therapeutic implications.
2. Excessive mechanistic density
While mechanistic detail is commendable, several sections (e.g., 2.3–2.5) are encyclopedic and could be condensed or summarized in schematic tables to improve readability. The repetition of signaling intermediates (e.g., JAK/STAT modifiers, histone modifiers) detracts from the overall flow.
Recommendation:
Condense the description of canonical IFNAR–JAK–STAT signaling and summarize noncanonical and regulatory pathways in tabular format (e.g., “Positive and Negative Regulators of IFN-I Signaling”).
Response: Thank you for the helpful suggestion. We have streamlined Section 2.3 and Section 2.4 (concise summaries of canonical IFNAR–JAK–STAT and noncanonical signaling), and reorganized Section 2.5 into schematic tables to avoid repetition.
3. Limited discussion of clinical translation
The section on “Targeted Therapy” is brief relative to the extensive mechanistic background. Readers would benefit from a concise table summarizing:
IFN-I–targeted therapies (anifrolumab, sifalimumab, rontalizumab, etc.)
JAK inhibitors (baricitinib, tofacitinib, upadacitinib)
Associated clinical outcomes and adverse effects.
Response: Thank you for the suggestion. We expanded the “Targeted Therapy” section and added a concise table summarizing IFN-I–targeted agents (anifrolumab, sifalimumab, rontalizumab, QX006N), JAK/TYK2 inhibitors (baricitinib, filgotinib, upadacitinib, ruxolitinib, deucravacitinib), and key upstream modulators (e.g., HCQ, TLR7/8 antagonists, cenerimod).
4. Integration of cross-disease insights
The review addresses SLE, RA, and vasculitis separately but does not fully develop their shared IFN-I signatures and potential for comparative endotyping. A synthesis section comparing these conditions (perhaps titled “Common and Divergent IFN-I Mechanisms Across Autoimmune Diseases”) would strengthen the paper’s translational impact.
Response: We agree that the precise reason for uniformly high IFN-I across clinically distinct entities remains incompletely resolved. Nevertheless, current evidence supports a unifying framework in which heterogeneous upstream triggers converge on conserved nucleic-acid–sensing and IFNAR–JAK–STAT pathways, with disease-specific tissue niches determining where the interferon program is most apparent. We have added 2–4 sentences in the Introduction to articulate this cross-disease synthesis and to clarify how it informs biomarker selection and pathway-targeted therapy.
5. Citation and referencing
While most citations are recent, the paper occasionally over-relies on secondary reviews. Incorporating primary studies — especially those identifying IFN-I signatures and therapeutic targets — would increase rigor. Additionally, citations should be checked for completeness (journal, year, volume).
Response: Thank you for the suggestion. In the Targeted Therapy section, we added primary studies to complement the existing reviews.
6. Stylistic and editorial considerations
Some sentences are excessively long and could be split for readability.
The abstract could be made more concise by removing redundant phrases such as “context-dependent, biphasic IFN response governed by dose, timing, and tissue microenvironment.”
Minor grammatical edits are needed (e.g., consistent use of “IFN-I,” “IFN-Is,” or “Type I IFNs”).
Response: Thank you for the helpful editorial guidance. We revised the abstract to remove redundant phrasing for readability, and we standardized terminology across the manuscript.
Reviewer 2 Report
Comments and Suggestions for Authors
Over the past 70 years since its discovery, interferons (IFNs) have occupied a key place among the effectors of innate immunity and have confirmed their role as molecules that control antiviral reactions and trigger the activation of adaptive immunity. The variety of physiological functions of IFNs indicates their control and regulatory role in maintaining homeostasis. The IFN system is a rapid response system and one of the most important components of natural (innate) immunity, which largely determines the course and outcome of viral infections. Decoding the signaling pathways of IFN formation and action has led to an understanding of the complexity and layering of regulatory mechanisms that collectively form the two-phase IFN response. However, prolonged activation of the IFN system suppresses the antiviral response and innate immunity, leading to chronic infection. Subsequent dysregulation of interferon type I (IFN-I) may underlie various autoimmune diseases.
The authors of this review analyze data indicating the role of dysregulation of IFN-I activity in systemic autoimmune diseases, including systemic lupus erythematosus (SLE), rheumatoid arthritis (RA), and various forms of vasculitis. In their opinion, these facts have served as a catalyst for intensive efforts to therapeutically modulate the IFN pathway. The paper presents data on the biology of IFN-I and the architecture of its signaling pathways, in which control points for regulating IFN-I activity are identified, and its functional landscape in the immune system is described. The data on the role of IFN-I in SLE, RA, and vasculitis are analyzed in separate sections, as well as therapeutic aspects and stratification strategies.
The last section "Therapeutic aspects and stratification strategies" should be noted, which discusses drugs that can be used to correct IFN-I signaling pathways, which is very important in clinical practice.
The authors believe that "therapeutic strategies targeting the IFN-I axis can mitigate abnormal inflammation; however, their wider application is limited by an incomplete description of the mechanisms, a limited amount of high-quality preclinical and clinical data, and the complexity of selectively modulating inflammation without compromising the body's defenses."
The review is written in understandable language for immunologists dealing with the problem of interferons and cytokines, but it is quite difficult for non-specialists. Therefore, it is necessary to provide a separate list of terms and abbreviations at the end or beginning of the review to facilitate the perception of the presented material.
The review certainly deserves the attention of immunologists. The conclusions made by the authors follows logically from the problem under discussion. The authors consider it necessary to continue research on the precise identification of IFN-I regulatory networks depending on the type of disease, as well as "refinement and validation of biomarker panels for stratification and monitoring of patients."
The cited sources are confirmed by recent publications and are relevant.
The review can be published in the Biomolecules journal.
Author Response
Comments to be transmitted to the Author
1. it is necessary to provide a separate list of terms and abbreviations at the end or beginning of the review to facilitate the perception of the presented material.
Response: Thank you for the thorough and constructive assessment. We have added a dedicated Glossary near the end of the manuscript (after the Conclusions).
Reviewer 3 Report
Comments and Suggestions for Authors
The manuscript “I Interferon Axis in Systemic Autoimmune Diseases: From Molecular Pathways to Targeted Therapy”, submitted to the journal Medicina, is devoted to the description the IFN-I signaling axis, elucidating the contributions of IFN-Is to the pathogenesis of systemic lupus erythematosus , rheumatoid arthritis, and vasculitis, and therapeutic strategies that directly or indirectly target the IFN-I system. The review covers a broad spectrum of Type I Interferon axis not only in systemic autoimmune diseases but also in the innate immunity. The theme is of importance for the improvement the effectiveness of treatment patients with systemic lupus erythematosus, rheumatoid arthritis, and vasculitis.
I have some comments:
- Lines 39-42:
“Over the past five decades, converging evidence has implicated dysregulated IFN-Is
activity as a principal pathogenic driver in systemic autoimmune diseases, including systemic lupus erythematosus (SLE), rheumatoid arthritis (RA), and various forms of vasculitis [4-6].”
Comment:
The links provided do not contain articles on systemic lupus erythematosus.
- Lines 44-46:
“In RA, IFN-Is activation is less pronounced than in SLE; nevertheless, a subset of patients displays an interferon-inducible molecular phenotype that prognosticates therapeutic response and disease trajectory [8].”
Comment:
The link provided does not contain information on systemic lupus erythematosus.
- Lines 44-46:
“The RLR family comprises three members—RIG-I, MDA5, and laboratory of genetics and physiology 2 (LGP2)—which serve as the principal cytosolic sensors of RNA and detect RNA viruses[52-55].”
Comment:
It is better to write:
“The RLR family comprises three members—RIG-I, MDA5, and LGP2 (protein named “laboratory of genetics and physiology 2”)—which serve as the principal cytosolic sensors of RNA and detect RNA viruses[52-55].”
- In the Figure 1 authors depicts that NOD2 detects viral single-stranded RNA (ssRNA), but there is no citation and information in the text. Please add in the text this information and cite the appropriate article.
- It is advisable to provide the chapters 2. 5 and 3 with diagrams or illustrations.
- The word "vasculitis" appears in two different spellings in the text: vasculitis and vasculitides. Please use one spelling throughout the text. The preferred option is “vasculitis”.
Author Response
Comments to be transmitted to the Author
1. Lines 39-42:
“Over the past five decades, converging evidence has implicated dysregulated IFN-Is activity as a principal pathogenic driver in systemic autoimmune diseases, including systemic lupus erythematosus (SLE), rheumatoid arthritis (RA), and various forms of vasculitis [4-6].”
Comment: The links provided do not contain articles on systemic lupus erythematosus.
Response: Thank you for the helpful comment. We agree the previous citations did not include an SLE-specific paper; we have added an appropriate SLE reference and revised the sentence accordingly:
Rönnblom, L.E.; Alm, G.V.; Oberg, K.E. Possible induction of systemic lupus erythematosus by interferon-alpha treatment in a patient with a malignant carcinoid tumour. J. Intern. Med. 1990, 227, 207–210. DOI:10.1111/j.1365-2796.1990.tb00144.x.
2. Lines 44-46:
“In RA, IFN-Is activation is less pronounced than in SLE; nevertheless, a subset of patients displays an interferon-inducible molecular phenotype that prognosticates therapeutic response and disease trajectory [8].”
Comment: The link provided does not contain information on systemic lupus erythematosus.
Response: Thank you for pointing this out. To avoid confusion and ensure disease-specific referencing, we removed the comparison to SLE and restricted the statement to RA with RA-specific citations.
3. Lines 44-46:
“The RLR family comprises three members—RIG-I, MDA5, and laboratory of genetics and physiology 2 (LGP2)—which serve as the principal cytosolic sensors of RNA and detect RNA viruses[52-55].”
Comment: It is better to write:
“The RLR family comprises three members—RIG-I, MDA5, and LGP2 (protein named “laboratory of genetics and physiology 2”)—which serve as the principal cytosolic sensors of RNA and detect RNA viruses[52-55].”
Response: Thank you for the suggestion. We have revised the sentence as recommended and now define LGP2 on first mention.
4. In the Figure 1 authors depicts that NOD2 detects viral single-stranded RNA (ssRNA), but there is no citation and information in the text. Please add in the text this information and cite the appropriate article.
Response: Thank you for the comment. We have added text describing NOD2 sensing of viral ssRNA and cited the appropriate primary articles.
5. It is advisable to provide the chapters 2. 5 and 3 with diagrams or illustrations.
Response: Thank you for the suggestion. To improve readability and visual navigation, we summarized Section 2.5 in tables.
6. The word "vasculitis" appears in two different spellings in the text: vasculitis and vasculitides. Please use one spelling throughout the text. The preferred option is “vasculitis”.
Response: Thank you for pointing this out. We have standardized the terminology to “vasculitis” throughout the manuscript.
Round 2
Reviewer 1 Report
Comments and Suggestions for Authors
The authors have made significant improvements to the manuscript in response to the reviewer's comments. The revised version is clearer, better organized, and the additional details provided have enhanced both the scientific depth and readability of the paper. The authors have addressed all previous concerns adequately, and the current version is acceptable for publication in its present form.